# On Mixtures of Markov Chains

**Rishi Gupta**[*]
Stanford University
Stanford, CA 94305
rishig@cs.stanford.edu

**Ravi Kumar**
Google Research
Mountain View, CA 94043
ravi.k53@gmail.com

**Sergei Vassilvitskii**
Google Research
New York, NY 10011
sergeiv@google.com

## Abstract

We study the problem of reconstructing a mixture of Markov chains from the trajectories generated by random walks through the state space. Under mild non-degeneracy conditions, we show that we can uniquely reconstruct the underlying chains by only considering trajectories of length *three*, which represent triples of states. Our algorithm is spectral in nature, and is easy to implement.

## 1 Introduction

Markov chains are a simple and incredibly rich tool for modeling, and act as a backbone in numerous applications—from Pagerank for web search to language modeling for machine translation. While the true nature of the underlying behavior is rarely Markovian [6], it is nevertheless often a good mathematical assumption.

In this paper, we consider the case where we are given observations from a *mixture of L Markov chains*, each on the same $n$ states, with $n \geq 2L$. Each observation is a series of states, and is generated as follows: a Markov chain and starting state are selected from a distribution $\mathcal{S}$, and then the selected Markov chain is followed for some number of steps. The goal is to recover $\mathcal{S}$ and the transition matrices of the $L$ Markov chains from the observations.

When all of the observations follow from a single Markov chain (namely, when $L = 1$), recovering the mixture parameters is easy. A simple calculation shows that the empirical starting distribution and the empirical transition probabilities form the maximum likelihood Markov chain. So we are largely interested in the case when $L > 1$.

As a motivating example, consider the usage of a standard maps app on a phone. There are a number of different reasons one might use the app: to search for a nearby business, to get directions from one point to another, or just to orient oneself. However, the users of the app never specify an explicit intent, rather they swipe, type, zoom, etc., until they are satisfied. Each one of the latent intents can be modeled by a Markov chain on a small state space of actions. If the assignment of each session to an intent were explicit, recovering these Markov chains would simply reduce to several instances of the $L = 1$ case. Here we are interested in the unsupervised setting of finding the underlying chains when this assignment is unknown. This allows for a better understanding of usage patterns. For example:

- Common uses for the app that the designers had not expected, or had not expected to be common. For instance, maybe a good fraction of users (or user sessions) simply use the app to check the traffic.
- Whether different types of users use the app differently. For instance, experienced users might use the app differently than first time users, either due to having different goals, or due to accomplishing the same tasks more efficiently.
- Undiscoverable flows, with users ignoring a simple, but hidden menu setting, and instead using a convoluted path to accomplish the same goal.

---

[*]Part of this work was done while the author was visiting Google Research.

The question of untangling mixture models has received a lot of attention in a variety of different situations, particularly in the case of learning mixtures of Gaussians, see for example the seminal work of [8], as well as later work by [5, 11, 15] and the references therein. This is, to the best of our knowledge, the first work that looks at unraveling mixtures of Markov chains.

There are two immediate approaches to solving this problem. The first is to use the Expectation-Maximization (EM) algorithm [9]. The EM algorithm starts by guessing an initial set of parameters for the mixture, and then performs local improvements that increase the likelihood of the proposed solution. The EM algorithm is a useful benchmark and will converge to some local optimum, but it may be slow to get there [12], and has are no guarantees on the quality of the final solution.

The second approach is to model the problem as a Hidden Markov Model (HMM), and employ the machinery for learning HMMs, particularly the recent tensor decomposition methods [2, 3, 10]. As in our case, this machinery relies on having more observed states than hidden states. Unfortunately, directly modeling a Markov chain mixture as an HMM (or as a mixture of HMMs, as in [13]) requires $nL$ hidden states for $n$ observed states. Given that, one could try adapting the tensor decomposition arguments from [3] to our problem, which is done in Section 4.3 of [14]. However, as the authors note, this requires accurate estimates for the distribution of trajectories (or trails) of length five, whereas our results only require estimates for the distribution of trails of length three. This is a large difference in the amount of data one might need to collect, as one would expect to need $\Theta(n^t)$ samples to estimate the distribution of trails of length $t$.

An entirely different approach is to assume a Dirichlet prior on the mixture, and model the problem as learning a mixture of Dirichlet distributions [14]. Besides requiring the Dirichlet prior, this method also requires very long trails. Finally, we would like to note a connection to the generic identifiability results for HMMs and various mixture models in [1]. Their results are existential rather than algorithmic, but dimension three also plays a central role.

**Our contributions.** We propose and study the problem of reconstructing a mixture of Markov chains from a set of observations, or trajectories. Let a $t$-trail be a trajectory of length $t$: a starting state chosen according to $\mathcal{S}$ along with $t - 1$ steps along the appropriate Markov chain.

(i) We identify a weak non-degeneracy condition on mixtures of Markov chains and show that under that non-degeneracy condition, 3-trails are sufficient for recovering the underlying mixture parameters. We prove that for random instances, the non-degeneracy condition holds with probability 1.

(ii) Under the non-degeneracy condition, we give an efficient algorithm for uniquely recovering the mixture parameters given the exact distribution of 3-trails.

(iii) We show that our algorithm outperforms the most natural EM algorithm for the problem in some regimes, despite EM being orders of magnitude slower.

**Organization.** In Section 2 we present the necessary background material that will be used in the rest of the paper. In Section 3 we state and motivate the non-degeneracy condition that is sufficient for unique reconstruction. Using this assumption, in Section 4 we present our four-step algorithm for reconstruction. In Section 5 we present our experimental results on synthetic and real data. In Section 6 we show that random instances are non-degenerate with probability 1.

## 2  Preliminaries

Let $[n] = \{1, \ldots, n\}$ be a state space. We consider Markov chains defined on $[n]$. For a Markov chain given by its $n \times n$ transition matrix $M$, let $M(i, j)$ denote the probability of moving from state $i$ to state $j$. By definition, $M$ is a stochastic matrix, $M(i, j) \geq 0$ and $\sum_j M(i, j) = 1$. (In general we use $A(i, j)$ to denote the $(i, j)$th entry of a matrix $A$.)

For a matrix $A$, let $\overline{A}$ denote its transpose. Every $n \times n$ matrix $A$ of rank $r$ admits a singular value decomposition (SVD) of the form $A = U\Sigma\overline{V}$ where $U$ and $V$ are $n \times r$ orthogonal matrices and $\Sigma$ is an $r \times r$ diagonal matrix with non-negative entries. For an $L \times n$ matrix $B$ of full rank, its *right pseudoinverse* $B^{-1}$ is an $n \times L$ matrix of full rank such that $BB^{-1} = I$; it is a standard fact that pseudoinverses exist and can be computed efficiently when $n \geq L$.

We now formally define a *mixture of Markov chains* $(\mathcal{M}, \mathcal{S})$. Let $L \geq 1$ be an integer. Let $\mathcal{M} = \{M^1, \ldots, M^L\}$ be $L$ transition matrices, all defined on $[n]$. Let $\mathcal{S} = \{s^1, \ldots, s^L\}$ be a

corresponding set of positive $n$-dimensional vectors of *starting probabilities* such that $\sum_{\ell,i} s_i^\ell = 1$. Given $\mathcal{M}$ and $\mathcal{S}$, a $t$-trail is generated as follows: first pick the chain $\ell$ and the starting state $i$ with probability $s_i^\ell$, and then perform a random walk according to the transition matrix $M^\ell$, starting from $i$, for $t-1$ steps.

Throughout, we use $i, j, k$ to denote states in $[n]$ and $\ell$ to denote a particular chain. Let $1_n$ be a column vector of $n$ 1's.

**Definition 1** (Reconstructing a Mixture of Markov Chains). *Given a (large enough) set of trails generated by a mixture of Markov chains and an $L > 1$, find the parameters $\mathcal{M}$ and $\mathcal{S}$ of the mixture.*

Note that the number of parameters is $O(n^2 \cdot L)$. In this paper, we focus on a seemingly restricted version of the reconstruction problem, where all of the given trails are of length three, i.e., every trail is of the form $i \to j \to k$ for some three states $i, j, k \in [n]$. Surprisingly, we show that 3-trails are sufficient for perfect reconstruction.

By the definition of mixtures, the probability of generating a given 3-trail $i \to j \to k$ is

$$\sum_\ell s_i^\ell \cdot M^\ell(i,j) \cdot M^\ell(j,k), \tag{1}$$

which captures the stochastic process of choosing a particular chain $\ell$ using $\mathcal{S}$ and taking two steps in $M^\ell$. Since we only observe the trails, the choice of the chain $\ell$ in the above process is latent. For each $j \in [n]$, let $O_j$ be an $n \times n$ matrix such that $O_j(i,k)$ equals the value in (1). It is easy to see that using $O((n^3 \log n)/\epsilon^2)$ sample trails, every entry in $O_j$ for every $j$ is approximated to within an additive $\pm\epsilon$. For the rest of the paper, we assume we know each $O_j(i,k)$ exactly, rather than an approximation of it from samples.

We now give a simple decomposition of $O_j$ in terms of the transition matrices in $\mathcal{M}$ and the starting probabilities in $\mathcal{S}$. Let $P_j$ be the $L \times n$ matrix whose $(\ell, i)$th entry denotes the probability of using chain $\ell$, starting in state $i$, and transitioning to state $j$, i.e., $P_j(\ell, i) = s_i^\ell \cdot M^\ell(i,j)$. In a similar manner, let $Q_j$ be the $L \times n$ matrix whose $(\ell, k)$th entry denotes the probability of starting in state $j$, and transitioning to state $k$ under chain $\ell$, i.e., $Q_j(\ell, k) = s_j^\ell \cdot M^\ell(j,k)$. Finally, let $S_j = \mathrm{diag}(s_j^1, \ldots, s_j^L)$ be the $L \times L$ diagonal matrix of starting probabilities in state $j$. Then,

$$O_j = \overline{P_j} \cdot S_j^{-1} \cdot Q_j. \tag{2}$$

This decomposition will form the key to our analysis.

## 3   Conditions for unique reconstruction

Before we delve into the details of the algorithm, we first identify a condition on the mixture $(\mathcal{M}, \mathcal{S})$ such that there is a unique solution to the reconstruction problem when we consider trails of length three. (To appreciate such a need, consider a mixture where two of the matrices $M^\ell$ and $M^{\ell'}$ in $\mathcal{M}$ are identical. Then for a fixed vector $v$, any $s^\ell$ and $s^{\ell'}$ with $s^\ell + s^{\ell'} = v$ will give the same observations, regardless of the length of the trails.) To motivate the condition we require, consider again the sets of $L \times n$ matrices $\mathcal{P} = \{P_1, \ldots, P_n\}$ and $\mathcal{Q} = \{Q_1, \ldots, Q_n\}$ as defined in (2). Together these matrices capture the $n^2 L - 1$ parameters of the problem, namely, $n-1$ for each of the $n$ rows of each of the $L$ transition matrices $M^\ell$, and $nL - 1$ parameters defining $\mathcal{S}$. However, together $\mathcal{P}$ and $\mathcal{Q}$ have $2n^2 L$ entries, implying algebraic dependencies between them.

**Definition 2** (Shuffle pairs). *Two ordered sets $\mathcal{X} = \{X_1, \ldots, X_n\}$ and $\mathcal{Y} = \{Y_1, \ldots, Y_n\}$ of $L \times n$ matrices are* shuffle pairs *if the $j$th column of $X_i$ is identical to the $i$th column of $Y_j$ for all $i, j \in [n]$.*

Note that $\mathcal{P}$ and $\mathcal{Q}$ are shuffle pairs. We state an equivalent way of specifying this definition. Consider a $2nL \times n^2$ matrix $\mathcal{A}(\mathcal{P}, \mathcal{Q})$ that consists of a top and a bottom half. The top half is an $nL \times n^2$ block diagonal matrix with $P_i$ as the $i$th block. The bottom half is a concatenation of $n$ different $nL \times n$ block diagonal matrices; the $i$th block of the $j$th matrix is the $j$th column of $-Q_i$. A representation of $\mathcal{A}$ is given in Figure 1. As intuition, note that in each column, the two blocks of $L$ entries are the same up to negation. Let $F$ be the $L \times 2nL$ matrix consisting of $2n$ $L \times L$ identity matrices in a row. It is straightforward to see that $\mathcal{P}$ and $\mathcal{Q}$ are shuffle pairs if and only if $F \cdot \mathcal{A}(\mathcal{P}, \mathcal{Q}) = 0$.

Let the *co-kernel* of a matrix $X$ be the vector space comprising the vectors $v$ for which $vX = 0$. We have the following definition.

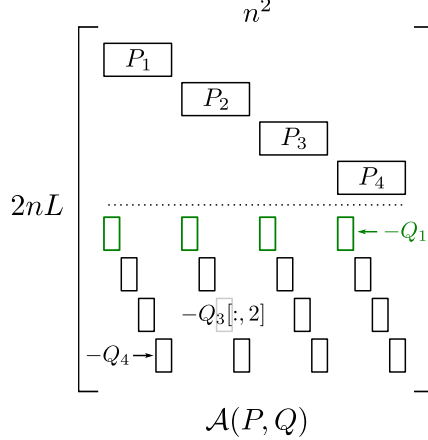

Figure 1: $\mathcal{A}(\mathcal{P}, \mathcal{Q})$ for $L = 2, n = 4$. When $\mathcal{P}$ and $\mathcal{Q}$ are shuffle pairs, each column has two copies of the same $L$-dimensional vector (up to negation). $\mathcal{M}$ is well-distributed if there are no non-trivial vectors $v$ for which $v \cdot \mathcal{A}(\mathcal{P}, \mathcal{Q}) = 0$.

**Definition 3** (Well-distributed). *The set of matrices $\mathcal{M}$ is* well-distributed *if the co-kernel of $\mathcal{A}(\mathcal{P}, \mathcal{Q})$ has rank $L$.*

Equivalently, $\mathcal{M}$ is well-distributed if the co-kernel of $\mathcal{A}(\mathcal{P}, \mathcal{Q})$ is spanned by the rows of $F$. Section 4 shows how to uniquely recover a mixture from the 3-trail probabilities $O_j$ when $\mathcal{M}$ is well-distributed and $\mathcal{S}$ has only non-zero entries. Section 6 shows that nearly all $\mathcal{M}$ are well-distributed, or more formally, that the set of non well-distributed $\mathcal{M}$ has (Lebesgue) measure 0.

## 4 Reconstruction algorithm

We present an algorithm to recover a mixture from its induced distribution on 3-trails. We assume for the rest of the section that $\mathcal{M}$ is well-distributed (see Definition 3) and $\mathcal{S}$ has only non-zero entries, which also means $P_j$, $Q_j$, and $O_j$ have rank $L$ for each $j$.

At a high level, the algorithm begins by performing an SVD of each $O_j$, thus recovering both $\overline{P_j}$ and $Q_j$, as in (2), up to unknown rotation and scaling. The key to undoing the rotation will be the fact that the sets of matrices $\mathcal{P}$ and $\mathcal{Q}$ are shuffle pairs, and hence have algebraic dependencies.

More specifically, our algorithm consists of four high-level steps. We first list the steps and provide an informal overview; later we will describe each step in full detail.

(i) *Matrix decomposition*: Using SVD, we compute a decomposition $O_j = \overline{U_j} \Sigma_j V_j$ and let $P'_j = U_j$ and $Q'_j = \Sigma_j V_j$. These are the initial guesses at $(\overline{P_j}, Q_j)$. We prove in Lemma 4 that there exist $L \times L$ matrices $Y_j$ and $Z_j$ so that $P_j = Y_j P'_j$ and $Q_j = Z_j Q'_j$ for each $j \in [n]$.

(ii) *Co-kernel*: Let $\mathcal{P}' = \{P'_1, \ldots, P'_n\}$, and $\mathcal{Q}' = \{Q'_1, \ldots, Q'_n\}$. We compute the co-kernel of matrix $\mathcal{A}(\mathcal{P}', \mathcal{Q}')$ as defined in Section 3, to obtain matrices $Y'_j$ and $Z'_j$. We prove that there is a single matrix $R$ for which $Y_j = RY'_j$ and $Z_j = RZ'_j$ for all $j$.

(iii) *Diagonalization*: Let $\overline{R'}$ be the matrix of eigenvectors of $(Z'_1 \overline{Y'_1})^{-1}(Z'_2 \overline{Y'_2})$. We prove that there is a permutation matrix $\Pi$ and a diagonal matrix $D$ such that $R = D\Pi R'$.

(iv) *Two-trail matching*: Given $O_j$ it is easy to compute the probability distribution of the mixture over 2-trails. We use these to solve for $D$, and using $D$, compute $R$, $Y_j$, $P_j$, and $S_j$ for each $j$.

### 4.1 Matrix decomposition

From the definition, both $P'_j$ and $Q'_j$ are $L \times n$ matrices of full rank. The following lemma states that the SVD of the product of two matrices $\overline{A}$ and $B$ returns the original matrices up to a change of basis.

**Lemma 4.** *Let $A, B, C, D$ be $L \times n$ matrices of full rank, such that $\overline{A}B = \overline{C}D$. Then there is an $L \times L$ matrix $X$ of full rank such that $C = X^{-1}A$ and $D = \overline{X}B$.*

*Proof.* Note that $\overline{A} = \overline{A}BB^{-1} = \overline{C}DB^{-1} = \overline{C}W$ for $W = DB^{-1}$. Since $A$ has full rank, $W$ must as well. We then get $\overline{C}D = \overline{A}B = \overline{C}WB$, and since $\overline{C}$ has full column rank, $D = WB$. Setting $X = \overline{W}$ completes the proof. $\square$

Since $O_j = \overline{P_j}(S_j^{-1}Q_j)$ and $O_j = \overline{P_j'}Q_j'$, Lemma 4 implies that there exists an $L \times L$ matrix $X_j$ of full rank such that $P_j = X_j^{-1}P_j'$ and $Q_j = S_j\overline{X_j}Q_j'$. Let $Y_j = X_j^{-1}$, and let $Z_j = S_j\overline{X_j}$. Note that both $Y_j$ and $Z_j$ have full rank, for each $j$. Once we have $Y_j$ and $Z_j$, we can easily compute both $P_j$ and $S_j$, so we have reduced our problem to finding $Y_j$ and $Z_j$.

### 4.2 Co-kernel

Since $(\mathcal{P}, \mathcal{Q})$ is a shuffle pair, $((Y_jP_j')_{j\in[n]}, (Z_jQ_j')_{j\in[n]})$ is also a shuffle pair. We can write the latter fact as $\mathcal{B}(Y, Z)\,\mathcal{A}(P', Q') = 0$, where $\mathcal{B}(Y, Z)$ is the $L \times 2nL$ matrix comprising $2n$ matrices concatenated together; first $Y_j$ for each $j$, and then $Z_j$ for each $j$. We know $\mathcal{A}(P', Q')$ from the matrix decomposition step, and we are trying to find $\mathcal{B}(Y, Z)$. By well-distributedness, the co-kernel of $\mathcal{A}(P, Q)$ has rank $L$. Let $D$ be the $2nL \times 2nL$ block diagonal matrix with the diagonal entries $(Y_1^{-1}, Y_2^{-1}, \ldots, Y_n^{-1}, Z_1^{-1}, Z_2^{-1}, \ldots, Z_n^{-1})$. Then $\mathcal{A}(P', Q') = D\,\mathcal{A}(P, Q)$. Since $D$ has full rank, the co-kernel of $\mathcal{A}(P', Q')$ has rank $L$ as well.

We compute an arbitrary basis of the co-kernel of $\mathcal{A}(P', Q'))$,[2] and write it as an $L \times 2nL$ matrix as an initial guess $\mathcal{B}(Y', Z')$ for $\mathcal{B}(Y, Z)$. Since $\mathcal{B}(Y, Z)$ lies in the co-kernel of $\mathcal{A}(P', Q')$, and has exactly $L$ rows, there exists an $L \times L$ matrix $R$ such that $B(Y, Z) = R\,B(Y', Z')$, or equivalently, such that $Y_j = RY_j'$ and $Z_j = RZ_j'$ for every $j$. Since $Y_j$ and $Z_j$ have full rank, so does $R$. Now our problem is reduced to computing $R$.

### 4.3 Diagonalization

Recall from the matrix decomposition step that there exist matrices $X_j$ such that $Y_j = X_j^{-1}$ and $Z_j = S_j\overline{X_j}$. Hence $Z_j'\overline{Y_j'} = (R^{-1}Z_j)(\overline{Y_j}\,\overline{R^{-1}}) = R^{-1}S_j\overline{R^{-1}}$. It seems difficult to compute $R$ directly from equations of the form $R^{-1}S_j\overline{R^{-1}}$, but we can multiply any two of them together to get, e.g., $(Z_1'\overline{Y_1'})^{-1}(Z_2'\overline{Y_2'}) = \overline{R}S_1^{-1}S_2\overline{R^{-1}}$.

Since $S_1^{-1}S_2$ is a diagonal matrix, we can diagonalize $\overline{R}S_1^{-1}S_2\overline{R^{-1}}$ as a step towards computing $R$. Let $\overline{R'}$ be the matrix of eigenvectors of $\overline{R}S_1^{-1}S_2\overline{R^{-1}}$. Now, $\overline{R}$ is determined up to a scaling and ordering of the eigenvectors. In other words, there is a permutation matrix $\Pi$ and diagonal matrix $D$ such that $R = D\Pi R'$.

### 4.4 Two-trail matching

First, $O_j1_n = \overline{P_j}S_j^{-1}Q_j1_n = \overline{P_j}1_L$ for each $j$, since each row of $S_j^{-1}Q_j$ is simply the set of transition probabilities out of a particular Markov chain and state. Another way to see it is that both $O_j1_n$ and $\overline{P_j}1_L$ are vectors whose $i$th coordinate is the probability of the trail $i \to j$.

From the first three steps of the algorithm, we also have $P_j = Y_jP_j' = RY_j'P_j' = D\Pi R'Y_j'P_j'$.

Hence $\overline{1_L}D\Pi = \overline{1_L}P_1(R'Y_1'P_1')^{-1} = \overline{O_11_n}(R'Y_1'P_1')^{-1}$, where the inverse is a pseudoinverse. We arbitrarily fix $\Pi$, from which we can compute $D$, $R$, $Y_j$, and finally $P_j$ for each $j$. From the diagonalization step (Section 4.3), we can also compute $S_j = R(Z_j'\overline{Y_j'})\overline{R}$ for each $j$.

Note that the algorithm implicitly includes a proof of uniqueness, up to a setting of $\Pi$. Different orderings of $\Pi$ correspond to different orderings of $M^\ell$ in $\mathcal{M}$.

# 5 Experiments

We have presented an algorithm for reconstructing a mixture of Markov chains from the observations, assuming the observation matrices are known exactly. In this section we demonstrate that the algorithm is efficient, and performs well even when we use empirical observations. In addition, we also compare its performance against the most natural EM algorithm for the reconstruction problem.

**Synthetic data.** We begin by generating well distributed instances $\mathcal{M}$ and $S$. Let $D_n$ be the uniform distribution over the $n$-dimensional unit simplex, namely, the uniform distribution over vectors in $\mathbb{R}^n$ whose coordinates are non-negative and sum to 1.

For a specific $n$ and $L$, we generate an instance $(\mathcal{M}, \mathcal{S})$ as follows. For each state $i$ and Markov chain $M^\ell$, the set of transition probabilities leaving $i$ is distributed as $D_n$. We draw each $s^\ell$ from $D_n$ as well, and then divide by $L$, so that the sum over all $s^\ell(i)$ is 1. In other words, each trail is equally likely to come from any of the $L$ Markov chains. This restriction has little effect on our algorithm, but is needed to make EM tractable. For each instance, we generate $T$ samples of 3-trails. The results that we report are the medians of 100 different runs.

**Metric for synthetic data.** Our goal is exact recovery of the underlying instance $\mathcal{M}$. Given two $n \times n$ matrices $A$ and $B$, the *error* is the average total variation distance between the transition probabilities: $\mathrm{error}(A, B) = 1/(2n) \cdot \sum_{i,j} |A(i,j) - B(i,j)|$. Given a pair of instances $\mathcal{M} = \{M^1, \dots, M^L\}$ and $\mathcal{N} = \{N^1, \dots, N^L\}$ on the same state space $[n]$, the *recovery error* is the minimum average error over all matchings of chains in $\mathcal{N}$ to $\mathcal{M}$. Let $\sigma$ be a permutation on $[L]$, then:

$$\text{recovery error}(\mathcal{M}, \mathcal{N}) = \min_\sigma \frac{1}{L} \sum_\ell \mathrm{error}(M^\ell, N^{\sigma(\ell)}).$$

Given all the pairwise errors $\mathrm{error}(M^\ell, N^p)$, this minimum can be computed in time $O(L^3)$ by the Hungarian algorithm. Note that the recovery error ranges from 0 to 1.

**Real data.** We use the last.fm 1K dataset[3], which contains the list of songs listened by heavy users of Last.Fm. We use the top 25 artist genres[4] as the states of the Markov chain. We consider the ten heaviest users in the data set, and for each user, consider the first 3001 state transitions that change their state. We break each sequence into 3000 3-trails. Each user naturally defines a Markov chain on the genres, and the goal is to recover these individual chains from the observed mixture of 3-trails.

**Metric for real data.** Given a 3-trail from one of the users, our goal is to predict which user the 3-trail came from. Specifically, given a 3-trail $t$ and a mixture of Markov chains $(\mathcal{M}, \mathcal{S})$, we assign $t$ to the Markov chain most likely to have generated it. A recovered mixture $(\mathcal{M}, \mathcal{S})$ thereby partitions the observed 3-trails into $L$ groups. The *prediction error* is the minimum over all matchings between groups and users of the fraction of trails that are matched to the wrong user. The prediction error ranges from 0 to $1 - 1/L$.

**Handling approximations.** Because the algorithm operates on real data, rather than perfect observation matrices, we make two minor modifications to make it more robust. First, in the diagonalization step (Section 4.3), we sum $(Z_i' \overline{Y_i})^{-1} (Z_{i+1}' \overline{Y_{i+1}})^{-1}$ over all $i$ before diagonalizing to estimate $R'$, instead of just using $i = 1$. Second, due to noise, the matrices $M$ that we recover at the end need not be stochastic. Following the work of [7] we normalize the values by first taking absolute values of all entries, and then normalizing so that each of the columns sums to 1.

**Baseline.** We turn to EM as a practical baseline for this reconstruction problem. In our implementation, we continue running EM until the log likelihood changes by less than $10^{-7}$ in each iteration; this corresponds to roughly 200-1000 iterations. Although EM continues to improve its solution past this point, even at the $10^{-7}$ cutoff, it is already 10-50x slower than the algorithm we propose.

## 5.1 Recovery and prediction error

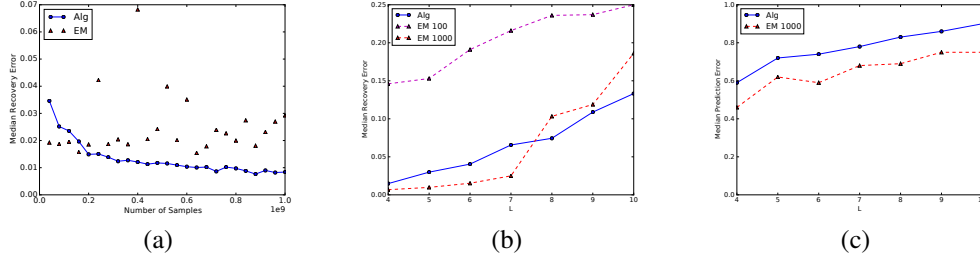

(a)               (b)               (c)

Figure 3: (a) Performance of EM and our algorithm vs number of samples (b) Performance of EM and our algorithm vs $L$ (synthetic data) (c) Performance of EM and our algorithm (real data)

For the synthetic data, we fix $n = 6$ and $L = 3$, and for each of the 100 instances generate a progressively larger set of samples. Recall that the number of unknown parameters grows as $\Theta(n^2 L)$, so even this relatively simple setting corresponds to over 100 unknown parameters. Figure 3(a) shows the median recovery error of both approaches. It is clear that the proposed method significantly outperforms the EM approach, routinely achieving errors of 10-90% lower. Furthermore, while we did not make significant attempts to speed up EM, it is already over 10x slower than our algorithm at $n = 6$ and $L = 3$, and becomes even slower as $n$ and $L$ grow.

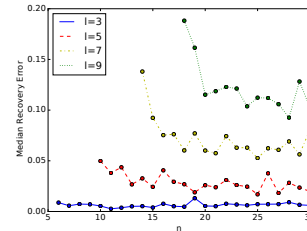

Figure 2: Performance of the algorithm as a function of $n$ and $L$ for a fixed number of samples.

In Figure 3(b) we study the error as a function of $L$. Our approach is significantly faster, and easily outperforms EM at 100 iterations. Running EM for 1000 iterations results with prediction error on par with our algorithm, but takes orders of magnitude more time to complete.

For the real data, there are $n = 25$ states, and we tried $L = 4, \dots, 10$ for the number of users. We run EM for 500 iterations and show the results in Figure 3(c). While our algorithm slightly underperforms EM, it is significantly faster in practice.

### 5.2 Dependence on $n$ and $L$

To investigate the dependence of our approach on the size of the input, namely $n$ and $L$, we fix the number of samples to $10^8$ but vary both the number of states from 6 to 30, as well as the number of chains from 3 to 9. Recall that the number of parameters grows as $n^2 L$, therefore, the largest examples have almost 1000 parameters that we are trying to fit.

We plot the results in Figure 2. As expected, the error grows linearly with the number of chains. This is expected — since we are keeping the number of samples fixed, the relative error (from the true observations) grows as well. It is therefore remarkable that the error grows only linearly with $L$.

We see more interesting behavior with respect to $n$. Recall that the proofs required $n \geq 2L$. Empirically we see that at $n = 2L$ the approach is relatively brittle, and errors are relatively high. However, as $n$ increases past that, we see the recovery error stabilizes. Explaining this behavior formally is an interesting open question.

## 6 Analysis

We now show that nearly all $\mathcal{M}$ are well-distributed (see Definition 3), or more formally, that the set of non well-distributed $\mathcal{M}$ has (Lebesgue) measure 0 for every $L > 1$ and $n \geq 2L$.

We first introduce some notation. All arrays and indices are 1-indexed. In previous sections, we have interpreted $i, j, k$, and $\ell$ as states or as indices of a mixture; in this section we drop these interpretations and just use them as generic indices.

For vectors $v_1, \ldots, v_n \in \mathbb{R}^L$, let $v_{[n]}$ denote $(v_1, \ldots, v_n)$, and let $*(v_1, \ldots, v_n)$ denote the $v_i$'s concatenated together to form a vector in $\mathbb{R}^{nL}$. Let $v_i[j]$ denote the $j$th coordinate of vector $v_i$.

We first show that there exists at least one well-distributed $\mathcal{P}$ for each $n$ and $L$.

**Lemma 5** (Existence of a well-distributed $\mathcal{P}$). *For every $n$ and $L$ with $n \geq 2L$, there exists a $\mathcal{P}$ for which the co-kernel of $\mathcal{A}(\mathcal{P}, \mathcal{Q})$ has rank $L$.*

*Proof.* It is sufficient to show it for $n = 2L$, since for larger $n$ we can pad with zeros. Also, recall that $F \cdot \mathcal{A}(\mathcal{P}, \mathcal{Q}) = 0$ for any $\mathcal{P}$, where $F$ is the $L \times 2nL$ matrix consisting of $2n$ identity matrices concatenated together. So the co-kernel of any $\mathcal{A}(\mathcal{P}, \mathcal{Q})$ has rank at least $L$, and we just need to show that there exists a $\mathcal{P}$ where the co-kernel of $\mathcal{A}(\mathcal{P}, \mathcal{Q})$ has rank at most $L$.

Now, let $e_\ell$ be the $\ell$th basis vector in $\mathbb{R}^L$. Let $\mathcal{P}^* = (P_1^*, \ldots, P_n^*)$, and let $p_{ij}^*$ denote the $j$th column of $P_i^*$. We set $p_{ij}^*$ to the $(i, j)$th entry of

$$
\begin{pmatrix}
e_1 & e_2 & \cdots & e_L & e_1 & e_2 & \cdots & e_L \\
e_L & e_1 & \cdots & e_{L-1} & e_L & e_1 & \cdots & e_{L-1} \\
\vdots & & & \vdots & \vdots & & & \vdots \\
e_2 & e_3 & \cdots & e_1 & e_2 & e_3 & \cdots & e_1 \\
e_1 & e_2 & \cdots & e_L & e_L & e_1 & \cdots & e_{L-1} \\
e_L & e_1 & \cdots & e_{L-1} & e_{L-1} & e_L & \cdots & e_{L-2} \\
\vdots & & & \vdots & \vdots & & & \vdots \\
e_2 & e_3 & \cdots & e_1 & e_1 & e_2 & \cdots & e_L
\end{pmatrix}.
$$

Formally, $p_{ij}^* = \begin{cases} e_{j-i+1} & \text{if } i \leq L \text{ or } j \leq L \\ e_{j-i}, & \text{if } i, j > L \end{cases}$, where subscripts are taken mod $L$. Note that we can split the above matrix into four $L \times L$ blocks $\begin{pmatrix} E & E \\ E & E' \end{pmatrix}$ where $E'$ is a horizontal "rotation" of $E$.

Now, let $a_{[n]}, b_{[n]}$ be any vectors in $\mathbb{R}^L$ such that $v = *(a_1, \ldots, a_n, b_1, \ldots, b_n) \in \mathbb{R}^{2nL}$ is in the co-kernel of $\mathcal{A}(\mathcal{P}^*, \mathcal{Q}^*)$. Recall this means $v \cdot \mathcal{A}(\mathcal{P}^*, \mathcal{Q}^*) = 0$. Writing out the matrix $\mathcal{A}$, it is not too hard to see that this holds if and only if $\langle a_i, p_{ij}^* \rangle = \langle b_j, p_{ij}^* \rangle$ for each $i$ and $j$.

Consider the $i$ and $j$ where $p_{ij}^* = e_1$. For each $k \in [L]$, we have $a_k[1] = b_k[1]$ from the upper left quadrant, $a_k[1] = b_{L+k}[1]$ from the upper right quadrant, $a_{L+k}[1] = b_k[1]$ from the lower left quadrant, and $a_{L+k}[1] = b_{L+(k+1 \pmod{L})}[1]$ from the lower right quadrant. It is easy to see that these combine to imply that $a_i[1] = b_j[1]$ for all $i, j \in [n]$.

A similar argument for each $l \in [L]$ shows that $a_i[l] = b_j[l]$ for all $i, j$ and $l$. Equivalently, $a_i = b_j$ for each $i$ and $j$, which means that $v$ lives in a subspace of dimension $L$, as desired. $\square$

We now bootstrap from our one example to show that almost all $\mathcal{P}$ are well-distributed.

**Theorem 6** (Almost all $\mathcal{P}$ are well-distributed). *The set of non-well-distributed $\mathcal{P}$ has Lebesgue measure 0 for every $n$ and $L$ with $n \geq 2L$.*

*Proof.* Let $\mathcal{A}'(\mathcal{P}, \mathcal{Q})$ be all but the last $L$ rows of $\mathcal{A}(\mathcal{P}, \mathcal{Q})$. For any $\mathcal{P}$, let $h(\mathcal{P}) = \det |\mathcal{A}'(\mathcal{P}, \mathcal{Q}) \overline{\mathcal{A}'(\mathcal{P}, \mathcal{Q})}|$. Note that $h(\mathcal{P})$ is non-zero if and only if $\mathcal{P}$ is well-distributed. Let $\mathcal{P}^*$ be the $\mathcal{P}^*$ from Lemma 5. Since $\mathcal{A}'(\mathcal{P}^*, \mathcal{Q}^*)$ has full row rank, $h(\mathcal{P}^*) \neq 0$. Since $h$ is a polynomial function of the entries of $\mathcal{P}$, and $h$ is non-zero somewhere, $h$ is non-zero almost everywhere [4]. $\square$

## 7 Conclusions

In this paper we considered the problem of reconstructing Markov chain mixtures from given observation trails. We showed that unique reconstruction is algorithmically possible under a mild technical condition on the "well-separatedness" of the chains. While our condition is sufficient, we conjecture it is also necessary; proving this is an interesting research direction. Extending our analysis to work for the noisy case is also a plausible research direction, though we believe the corresponding analysis could be quite challenging.

## Footnotes

[2]For instance, by taking the SVD of $\mathcal{A}(P', Q')$, and looking at the singular vectors.

[3] http://mtg.upf.edu/static/datasets/last.fm/lastfm-dataset-1K.tar.gz

[4] http://static.echonest.com/Lastfm-ArtistTags2007.tar.gz

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
