[Reviews · NeurIPS 2016]

Reviewer 1

Summary

The paper proposes a novel approach to estimate the parameters of a mixture of Markov chains. It is based on samples of trajectories with length 3 (the 3-trails) of the finite mixture and uses SVD decompositions and simple linear algebra techniques to recover the parameters from the empirical distribution of these 3-trails. Some synthetic experiments and a real data application are provided.

Qualitative Assessment

The paper is globally sound and makes a new contribution to an important topic. However some technicalities need to be addressed and a revised version should be encouraged. Major remarks: - There is a confusion on whether the Markov chains under consideration are supposed to be stationary or not. Indeed, the concept of t-trail either requires that the Markov chains under study are stationary or one should specify that all the trails start with same initial distribution, i.e. these trails are observations of (X_1,X_2,X_3) and not (X_s, X_{s+1},X_{s+2}) for some s. I first understood that you adopt the second approach (as you count the parameters of initial distributions as free parameters) but in the real data experiments, you take many (3001)-trails and break them into 3000 overlapping 3-trails (by the way, you need a 3002-trail to obtain these). This causes two different problems: the first is related to stationarity and the second will be developed below. So, by breaking a t-trail into several 3-trails you implicitly assume that the chains follow a stationary regime. This should be stated more clearly. And this should be taken into account when discussing the number of parameters (the stationary distribution for the \ell-th Markov chain will be entirely determined by its transition M^\ell - as soon as some ergodicity conditions are satisfied on M^\ell). To summarize, breaking a trail into several t-trails only makes sense in a stationary regime. - The second problem induced by breaking a t-trail into several 3-trails lies in the overlap. One could safely consider non-overlapping 3-trails (resulting in a smaller amount of data). Here, the authors choose to rely on overlapping t-trails (at least this is done in the real data experiments). Then, the law of large numbers is satisfied for each Markov chain provided some ergodicity conditions are assumed on their transition. In particular, the statement that ‘It is easy to see that using O(1/\epsilon^2 log n) sample trails, every entry in O_j for every j is approximated to within an additive +/- \epsilon’ (lines 97-99) is not valid in general when using non independent samples. - In the synthetic experiments, computation of the error requires to explore all the permutations for matching \mathcal{M} with \mathcal{N}. This is not a trivial task. How do you proceed? (same question applies in the real data experiments, especially when you have L=10 groups). - In the real data experiments with L=10 groups, you assume that the group structure is known and given by the 10 different users. How did you compute the prediction error for L less than 10? What is the truth here? In the case of 10 groups, why not simply computing an adjusted rand index (ARI) to evaluate the classifications (yours and EM) with respect to ‘truth’? - Synthetic experiments rely on only 11 different runs which is really small. You should go to at least 100 replicates of your experiments for each sample size. - Synthetic experiments: the authors draw initial distributions s^\ell from the Dirichlet distribution on the simplex (indeed, uniform distribution on the simplex is the Dirichlet distribution with parameters (1,…,1)) and then divide by L to obtain a distribution on the pair (group, initial state). This boils down to assuming that all the groups have equal probability, which could be noticed. More importantly, what happens when the groups are not equally distributed? - Those results are reminiscent of results obtained by Allman et al. ‘Identifiability of parameters in latent structure models with many observed variables’, Annals of Statistics, 2009. In this paper, the authors rely on a result by Kruskal on 3-ways arrays to prove identifiability of different classes of model. Dimension 3 plays an important role and they also obtain ‘generic identifiability’ results meaning that the set of non-identifiable parameters has Lebesgue measure 0. I believe this reference could be included. Minor remarks: - Line 82, M^1, …, M^L are more precisely transition matrices of Markov chains (and not Markov chains by themselves). - Line 110, ‘To appreciate such a need, consider the obviously necessary, though not sufficient, condition that the matrices in M be distinct.’ I don’t see how this is related to the previous sentence. - Number of parameters (lines 113-114): it seems to me that the set \mathcal{S} corresponds to distributions on the pair (group, initial state), in which case, there are additional constraints on those parameters and they should reduce to L(n-1) free parameters (each one of the L Markov chains has an initial distribution determined by n-1 parameters). In any case, I mentioned above that the general setting of the paper should be the stationary regime in which case initial distribution is not a free parameter anymore. - Line 228: ‘we consider the first 3001 state transitions that change their state’: what do you mean exactly by ‘that change their state’? Do you mean that you discard data where X_s=X_{s+1}? If yes, why do you do that and why would this be valid? - line 236:’The prediction error ranges from 1/L to 1’/ I don’t understand that statement. I would say that it ranges from 0 to 1. - Baseline EM: how do you initialize the algorithm? How many starting points do you use? Typos: - Line 83, initial probability is denoted s^\ell(i) while later it becomes s^\ell_i. - Equation (2): I believe the last factor 2 should be removed. - Line 176, ‘(P,Q) is a shuffle pair’ should be (\mathcal{P},\mathcal{Q}). - Line 157: matrix R^’ should be R (both instances). - Line 223-224: E_l is not defined. It is related to ‘error’ above.

Confidence in this Review

3-Expert (read the paper in detail, know the area, quite certain of my opinion)


Reviewer 2

Summary

The new model is interesting and captures some of the new challenges in machine learning. The main idea is to use a "semantic vector" to represent the labels, which allows infinitely many labels, and also labels that are semantically correlated with each other. The hypothesis is a function f that maps pairs of (x,\lambda) (data and semantic label vector) to +-1 (the true label). The new model is a natural generalization to the multi-label setting, and is more suitable for modern supervised learning applications. The paper gives a generalization bound for the new model based on the VC-dimension. The technique is quite straight-forward and similar to the previous generalization bounds. The new model is quite clean and it would not be surprising to see other generalization bounds also generalized to this new setting. One questionable point about the model is that it requires the semantic vectors for labels are known (and these are generalized independent of the training samples x). It would be interesting to see whether it is possible to learn both the semantic vectors and labeling function f using the same data set (information theoretically it seems possible given enough samples from each class). Or at least maybe given a few new samples in a new class, try to estimate the semantic vector for this new class. In experiments the paper uses word-embeddings which is reasonable, but being able to learn/adjust labels based on the data might also be interesting.

Qualitative Assessment

The new algorithm is based the observation that the backward matrix P and forward matrix Q are closely related (and both are closely related to the transition matrix of different chains). In particular, the two sets of matrices can be captured by the "shuffle set" property. This property allows the paper to show that a co-kernel matrix A(P,Q) is of low rank, which gives a way to relate the two rank L subspaces related to P and Q. The paper then uses standard approach of eigendecomposition to find a unique linear transformation to compute the matrices P and Q. The new observations of shuffle set property and co-kernel matrix are interesting techniques for spectral learning. The algorithm is clean and seem to perform well. The paper also shows the nondegeneracy condition is not restrictive. It would be interesting to see a more careful perturbation analysis, especially about the condition number of the co-kernel matrix. The analysis is mostly well-written, except in certain places the notations are a bit confusing. For example, Page 4, when describing step (ii), the result should be Y_j = R Y'_j and Z_j = R Z'_j (the ' appeared on incorrect matrices), and in (iii) it should be R = D\Pi \bar{R'}. Overall this is an interesting new result on spectral learning and contains some new techniques specially designed for mixture of Markov chains.

Confidence in this Review

2-Confident (read it all; understood it all reasonably well)


Reviewer 3

Summary

This paper tackles the problem of recovering a mixture of Markov chains from trajectories of length three. The authors give a condition under which the mixture is identifiable, and a spectral algorithm that recovers the parameters exactly in the noiseless case. The algorithm can easily be adapted to the noisy case, and experiments on synthetic and real data are provided.

Qualitative Assessment

This paper studies an interesting problem, with important applications. I am not very familiar with related work, and as such cannot accurately situate the problem in the context of other works, but the framework proposed by the authors appears to be new, and the technical contributions seem notable. I enjoyed reading this paper. Altogether I am in favor of accepting the paper. Here are the two main points that could be improved in my opinion: 1) I wonder if the definition of "well-distributedness" could be simplified (also see my question below.) Also I feel that proving the necessity of this condition is within reach, and would strengthen the paper. 2) the experimental results are a not very convincing. E.g. the regime chosen for the synthetic experiments (nb samples = 10^9, Ln^2 = 100) does not seem applicable to most realistic settings. A discussion on why EM seemingly outperforms the spectral approach when L increases would also be interesting. Questions: - l. 97, "It is easy to see that using...". That statement does not seem "easy" to me. At first sight, O(log n) samples seem just too few to estimate O(Ln^2) parameters. Can you make your statement more precise? What do you mean by approximated within +- eps? Also, when you say "sample trails", do you mean 3-trails? - Eq. 2: What is the factor 2 doing at the end? (probably a typo?) - The definition of well-distributedness is not trivial to fully grasp. Suppose the starting probability is uniform (i.e. s^l(i) = 1/nL), is the definition equivalent to saying "the set \mathcal{M} is linearly independent"? - l. 157: I suppose you mean $Y_j = R Y'_j$ and $Z_j = R Z'_j$ ? (note that the prime is at a different place.) - above l.224: by E_{\ell}[...] you probably mean the average? It is not immediately clear. Are you taking into account a distribution of the L components, e.g. as given by the starting probabilites? - Figure 3b: can you give an intuitive explanation on why EM becomes better when L is decreasing? Comments: - l.64: "the latter of which has no quality guarantees, even in practice". I don't understand what you mean by quality guarantees 'in practice'. - l. 114: I believe you mean 'row' instead of 'column' (the matrix is row-stochastic) - I think a discussion (or some examples of classes) of well-distributed vs non-well-distributed mixtures would be helpful. - l. 136: "to prove our bounds". Which bounds? I suppose you mean "to perfectly recover the mixture"? ===== UPDATE AFTER REBUTTAL Thank you for your rebuttal, which managed to answer some of my questions & comments above. One last comment: "With regard to independence, the ideal case would be if we had truly independent samples, eg, the same identifiable user coming back to last.FM every day." As a matter of fact, the dataset that you use *does* contain this. As each "play" event is timestamped, you could easily check if there was a long break between two successive plays.

Confidence in this Review

2-Confident (read it all; understood it all reasonably well)


Reviewer 4

Summary

This paper addresses the problem of reconstructing a mixture of Markov chain given observation trajectories. Authors it is sufficient to uniquely reconstruct the mixture of Markov chain given only the first three states of each trail and propose an efficient algorithm for that.

Qualitative Assessment

Since I am not very familiar with topic, I have some difficulty to understand the novel contribution and do not have valuable comments. From my point of view, it is not well motivated especially how authors come up with their reconstruction algorithm and why it works.

Confidence in this Review

1-Less confident (might not have understood significant parts)


Reviewer 5

Summary

This paper explores the problem of reconstructing mixtures of Markov chains by only considering triples of states. Specifically, they show that only the first three states of each trail need to be considered to recover mixture parameters. By recovering the Markov chain that produces a set of states, the authors can analyze user behavior. They show that their algorithm outperforms EM.

Qualitative Assessment

The paper gives a good background on the problem and a thorough explanation of the proof of their method. It would be nice to get an explanation of figure 3(a) and why the median recovery error of EM is all over the map compared to number of samples. They show good accuracy and performance compared to EM.

Confidence in this Review

1-Less confident (might not have understood significant parts)